# Multiple Instance Learning for Efficient Sequential Data Classification on Resource-constrained Devices

**Don Kurian Dennis**     **Chirag Pabbaraju**     **Harsha Vardhan Simhadri**     **Prateek Jain**
Microsoft Research, India
{t-dodenn, t-chpab, harshasi, prajain}@microsoft.com

## Abstract

We study the problem of fast and efficient classification of sequential data (such as time-series) on tiny devices, which is critical for various IoT related applications like audio keyword detection or gesture detection. Such tasks are cast as a standard classification task by sliding windows over the data stream to construct data points. Deploying such classification modules on tiny devices is challenging as predictions over sliding windows of data need to be invoked continuously at a high frequency. Each such predictor instance in itself is expensive as it evaluates large models over long windows of data. In this paper, we address this challenge by exploiting the following two observations about classification tasks arising in typical IoT related applications: (a) the "signature" of a particular class (e.g. an audio keyword) typically occupies a small fraction of the overall data, and (b) class signatures tend to be discernible early on in the data. We propose a method, EMI-RNN, that exploits these observations by using a multiple instance learning formulation along with an early prediction technique to learn a model that achieves better accuracy compared to baseline models, while simultaneously reducing computation by a large fraction. For instance, on a gesture detection benchmark [26], EMI-RNN requires 72x less computation than standard LSTM while improving accuracy by 1%. This enables us to deploy such models for continuous real-time prediction on devices as small as a Raspberry Pi0 and Arduino variants, a task that the baseline LSTM could not achieve. Finally, we also provide an analysis of our multiple instance learning algorithm in a simple setting and show that the proposed algorithm converges to the global optima at a linear rate, one of the first such result in this domain. The code for EMI-RNN is available at [14].

## 1 Introduction

**Classification of sequential data**: Several critical applications, especially in the Internet of Things (IoT) domain, require real-time predictions on sensor data. For example, wrist bands attempt to recognize gestures or activities (such as walking, climbing) from Inertial Measurement Unit (IMU) sensor data [2, 26]. Similarly, several audio applications require detection of specific keywords like "up", "down" or certain urban sounds [35, 32, 29, 5].

**LSTM based models**: Typically, such problems are modeled as a multi-class classification problem where each sliding window over the sensor data forms a sequential data point and each class represents one category of events to be detected. Additionally, there is a "noise" class which denotes a *no-event* data point. Recurrent Neural Networks (RNNs) like LSTMs [21] are a popular tool for modeling such problems where the RNN produces an *embedding* of the given sequential data point that can then be consumed by a standard classifier like logistic regression to predict label of the point [20, 30].

**Deployment on tiny devices**: Such event detection applications, especially in the IoT domain, require deployment of inference on tiny edge devices with capabilities comparable to an Arduino Uno [13] or a Raspberry Pi0 [19]. Further, the data is sampled at a high rate for good resolution

and the data buffer is often limited by small DRAMs. This means that the prediction needs to be completed before the buffer is refreshed. For example, the GesurePod device [26] requires the buffer to be refreshed every 100 ms and hence predictions must be completed in 100 ms using just 32KB of RAM. Another critical requirement is that the lag between the event and the detection should be very short. This not only requires fast but also "early" prediction, i.e., the solution must classify an event before even observing the entire event based on a prefix of the signature of the event.

**Drawbacks of existing solutions**: Unfortunately, existing state-of-the-art solutions like the ones based on standard RNN models like LSTM are difficult to deploy on tiny devices as they are computationally expensive. Typically, the training data is collected "loosely" to ensure that the event is captured *somewhere* in the time-series (see Figure 1). For example, in the case of the urban sound detection problem [5], the training data contains the relevant sound somewhere in the given 1-2 secs clip, while the actual vehicle noise itself might last only for few hundreds of milliseconds. That is, the sliding window length should be greater than 1 sec while ideally we should restrict ourselves to 100ms window lengths. But, it is difficult to pin-point the window of data where the particular event occurred. This phenomenon holds across various domains such as keyword detection [35], gesture recognition [26] and activity recognition [2]. Due to the large number of time-steps required to accommodate the buffer around the actual signature, the model size needs to be large enough to capture a large amount of variations in the location of the signature in the data point. Moreover, storing the entire data point can be challenging as the data for each time-step can be multidimensional, e.g., IMU data. Finally, due to the large computation time and the failure to tightly capture the signature of the event, the lag in prediction can be large.

**Our method**: In this work, we propose a method that addresses above mentioned challenges in deployment on tiny devices. In particular, our method is motivated by two observations: a) the actual "signature" of the event forms a small fraction of the data point. So, we can split the data point into multiple *instances* where only a few of the instances are *positive* instances, i.e., they contain the actual class/event while remaining instances represent noise (see Figure 1). b) The event class can be discerned by observing a prefix of the signature of the class. For example, if the audio category is that of repetitive sounds from a jack-hammer, then the class can be predicted with high accuracy using only a few initial samples from the data. We also observe that in many such applications, a few false positives are allowed as typically data from a positive event detection is uploaded to a powerful cloud server that can further prune out false positives.

Based on the above observations, we propose a robust multi-instance learning (MIL) method with early stopping. Our solution splits the given sequential data point into a collection of windows, referred to here as *instances* (see Figure 1). All the instances formed from the sequential data point are assigned it's class label. Based on observation (a) above, each sequential data typically has only a few consecutive *positive instances*, i.e., instances (sub-windows) that contain the true class "signature". The remaining *negative* instances have noisy labels and hence the classification problem with these labeled instances can be challenging. Our MIL algorithm uses the above mentioned structure with an iterative thresholding technique to provide an efficient and accurate algorithm. We train our model to predict the label after observing the data only for a few time-steps of the instance. That is, we jointly train our robust MIL model with an early classification loss function to obtain an accurate model that also reduces computation significantly by predicting early. For simplicity, we instantiate our algorithm for standard LSTM based architecture. However, our solution is independent of the base architecture and can be applied to other RNN based models like GRU [9], svdRNN [**?** ], UGRNN [11], CNN-LSTM [28] etc.

We also provide theoretical analysis of our thresholding based robust MIL method. Note that existing generic robust learning results [6, 15] do not apply to sequential data based MIL problems, as the number of incorrectly labeled instances form a large fraction of all instances (see Figure 1). Instead, we study our algorithm in a simple separable setting and show that despite presence of a large amount of noise, the algorithm converges to the optimal solution in a small number of iterations. Our analysis represents one of the first positive result for MIL in a more interesting *non-homogeneous* MIL setting that is known to be NP-hard in general [4, 27].

**Empirical results**: We present empirical results for our method when applied to five data sets related to activity detection, gesture recognition [2, 26], keyword detection [35] etc. For each of the benchmarks, we observe that our method is able to significantly reduce the computational time while also improving accuracy (in most of the benchmarks) over baseline LSTM architecture. In four of the five data sets we examine, our methods save $65 - 90\%$ of computation without losing accuracy

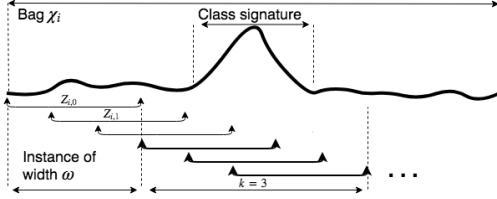

Figure 1: A time-series data point $X_i$ bagged into sliding windows of width $\omega$, just long enough to contain the true class signature.

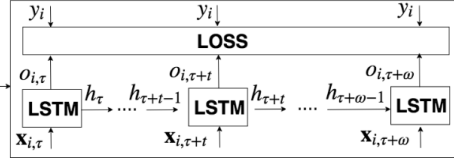

Figure 2: EMI-RNN architecture. $Z_{i,\tau}$ are the "positive" instances from each data point $X_i$ which are then combined with E-RNN loss (3.2.1) to train the model parameters.

compared to the baseline (LSTM trained on the entire time-series data) when compared at a fixed LSTM hidden dimension. In fact, for the keyword spotting task [35], our method can provide up to 2% more accurate predictions while saving $2-3\times$ in compute time. This enables deployment on a Raspberry Pi0 device, a task beyond the reach of normal LSTMs. Additionally, in all the data sets, our method is able to improve over baseline LSTM models using a much smaller hidden dimension, thus making inference up to 72x less expensive.

## 1.1 Related Work

**NN architectures for sequential data classification**: RNN based architectures are popular for learning with sequential data as they are able to capture long-term dependencies and tend to have a small working memory requirement. Gated architectures like LSTM [21], GRU [8, 10], UGRNN [11] have been shown to be successful for a variety of sequential data based applications. However, they tend to be more expensive than vanilla RNN, so there has been a lot of recent work on stabilizing training and performance of RNNs [3**?** ]. In addition, CNN based architectures [29] have also been proposed such as CNN-LSTM [16], but these methods tend to have a large working RAM requirement. Our method is independent of underlying architecture and can be combined with any of the above RNN-based techniques. For our experiments, we focus on the LSTM architecture as it tends to be stable and provides nearly state-of-the-art accuracy on most datasets.

**MIL/Robust learning**: Multi-instance learning (MIL) techniques are popular for modeling one-sided noise where positive bags of instances can have several negative instances as well. Existing practical algorithms [1] iteratively refine the set of positive instances to learn a robust classifier but typically do not have strong theoretical properties. On the other hand, [27] consider a homogeneous setting where one can obtain strong theoretical bounds but the assumptions are too strong for any practical setting and the resulting algorithm is just a standard classifier. In contrast, our algorithm provides significant accuracy improvements on benchmarks, and at the same time, we can analyze the algorithm in a *non-homogeneous* setting which represents one of the first positive results in this setting.

**Early classification**: Recently, several papers have studied the problem of early classification in sequential data [25, 33, 12, 36, 17], however these techniques assume a pre-trained classifier and learn a policy for early classification independently which can lead to significant loss in accuracy. [24] introduced an early-classification method for the specific task of human activity detection in video. Independent of our work, [7] developed an architecture to reduce computational cost of RNNs by skipping certain hidden-state updates. They show that this technique leads to almost no loss in accuracy. Their technique is complementary to our work as our focus is on early classification, i.e., predicting the class before observing the entire sequential data point. Further, our EMI-RNN architecture uses joint training with MIL formulation to ensure that such early classification is accurate for a large fraction of points.

## 2 Problem Formulation and Notation

Suppose we are given a dataset of labeled sequential data points $\mathcal{Z} = [(X_1, y_1), \ldots, (X_n, y_n)]$ where each *sequential* data point $X_i = [\mathbf{x}_{i1}, \mathbf{x}_{i2}, \ldots, \mathbf{x}_{iT}]$ is a sequence of $T$ data points with $\mathbf{x}_{i,t} \in \mathbb{R}^d$. That is, the $t$-th *time-step data point* of the $i$-th *sequential data point* is $\mathbf{x}_{i,t}$. Throughout the paper, $v_i$ denotes the $i$-th component of a vector $\mathbf{v}$. Let $y_i \in [L]$ denote the class of $i$-the data point $X_i$; $[L] := \{-1, 1, 2, \ldots, L\}$. Here $y_i = -1$ denotes "noisy" points that do not correspond to any class. For example, in keyword detection application, $X_i$ is an audio clip with $\mathbf{x}_{it}$ being the $t$-th sample from the clip and the $y_i$-th keyword spoken in the clip. If no keyword is present, $y_i = -1$.

Given $\mathcal{Z}$, the goal is to learn a classifier $f : \mathbb{R}^{d \times T} \to \mathbb{R}^L$ that can *accurately* predict the class of the input sequential data point $X$ using $\arg\max_b f_b(X)$. In addition, for $f$ to be deployable on tiny devices in real-world settings, it must satisfy **three key requirements**: a) small memory footprint, b) can be computed quickly on resource-constrained devices, c) should predict the correct class after observing as few *time-step* data points of $X$ as possible, i.e., as *early* as possible.

We focus on problems where each class has a certain "signature" in the sequential data (see Figure 1). The goal is to train $f$ to identify the signature efficiently and with minimal lag. Further, due to the architectural constraints of tiny devices, it might not be possible to buffer the entire sequential point in memory. This implies that early classification with small number of time-step data points is critical to minimizing the lag in prediction.

Due to the sequential nature of data, existing approaches use recurrent neural network (RNN) based solutions [21, 8]. However, as mentioned in Section 1, such solutions do not fare well on the three key requirements mentioned above due to training data inefficiency (see Section 1). That is, while the actual class-signature is a small part of the entire data point, in absence of more fine-grained labels, existing methods typically process the entire sequential data point to predict the class label which leads to expensive computation and large model sizes.

In the next section, we describe our method that addresses all the key requirements mentioned above — low memory footprint, low computational requirement and small lag in prediction.

## 3 Method

As mentioned in the previous section, although the entire sequential data point $X_i = [\mathbf{x}_{i1}, \mathbf{x}_{i2}, \ldots, \mathbf{x}_{iT}]$ is labeled with class $y_i$, there is a small subset of *consecutive* but *unknown* time-steps that capture the signature of the label, for instance the part of audio clip where a particular keyword is spoken (see Figure 1). Moreover, the label signature is significantly distinct from other labels, and hence can be detected reasonably early without even observing the entire signature. In this section, we introduce a method that exploits both these insights to obtain a classifier with smaller prediction complexity as well as smaller "lag" in label prediction.

We would like to stress that our method is orthogonal to the actual neural-network architecture used for classification and can be used with any of the state-of-the-art classification techniques employed for sequential data. In this work we instantiate our method with a simple LSTM based architecture for its ease of training [26, 18, 23]. Our method is based on two key techniques and next two subsections describe both the techniques separately; Section 3.3 then describes our final method that combines both the techniques to provide an efficient and "early" sequential data classifier.

### 3.1 MI-RNN

As the true label signature can be a tiny fraction of a given sequential data point, we break the data point itself into multiple overlapping instances such that at least one of the instance cover the true label signature (see Figure 1). That is, given a sequential data point $X_i$, we construct a bag $\mathcal{X}_i = [Z_{i,1}, \ldots, Z_{i,T-\omega+1}]$ where $Z_{i,\tau} = [\mathbf{x}_{i,\tau}, \ldots, \mathbf{x}_{i,\tau+\omega-1}] \in \mathbb{R}^{d \times \omega}$ represents the $\tau$-th instance (or sub-window) of data points. $\omega$ is the instance-length, i.e., the number of time-steps in each instance $Z_{i,\tau}$, and should be set close to the typical length of the label signature. For example, if keywords in our dataset can be all captured in 10 time-steps then $\omega$ should be set close to 10.

While a small number of $Z_{i,\tau}$ instances have label $y_i$, the remaining instances are essentially noise and should be set to have label $y_{i,\tau} = -1$. Since we do not have this label information apriori the label of *each* instance in the $i$-th sequential data point is initialized to $y_i$. This leads to a heavily noisy labelling. Because of this, the problem can be viewed as a multi-instance learning problem with only "bag" level label available rather than instance level labels. Similarly, the problem can also be viewed as a robust learning problem where the label for *most* of the instances is wrong.

Naturally, existing techniques for robust learning do not apply as they are not suitable for the large amount of noise in this setting. Moreover, these methods and multi-instance learning methods fail to exploit structure in the problem – the existence of consecutive set of instances with the given label. Instead, we study a simple optimization problem that captures the above mentioned structure:

$$\min_{f, s_i, 1 \le i \le n} \frac{1}{n} \sum_{i,\tau} \delta_{i,\tau} \ell(f(Z_{i,\tau}), y_i)), \text{ such that, } \delta_{i,\tau} = \begin{cases} 1, & \tau \in [s_i, s_i + k - 1], \\ 0, & \tau \notin [s_i, s_i + k - 1]. \end{cases} \quad (3.1.1)$$

**Algorithm 1** MI-RNN: Multi-instance RNN

**Require:** Sequential data: $\{(X_1, y_1), \ldots, (X_n, y_n)\}$ with T steps, instance-length $\omega$, parameter $k$
1: Create multiple instances: $\{Z_{i,\tau}, 1 \le \tau \le T - \omega + 1\}$ with $Z_{i,\tau} = [\mathbf{x}_{i,\tau}, \ldots, \mathbf{x}_{i,\tau+\omega-1}]$ for all $i$
2: Initialize: $f \leftarrow$ Train-LSTM($\{(Z_{i,\tau}, y_i), \forall i, \tau\}$)
3: **repeat**
4: $\quad f \leftarrow$ Train-LSTM($\{(Z_{i,\tau}, y_{i,\tau}), \tau \in [s_i, s_i + k - 1], \forall i\}$)
5: $\quad s_i = \arg\max_{\tau'} \sum_{\tau' \le \tau \le \tau'+k-1} f_{y_i}(Z_{i,\tau'}), \forall i$
6: **until** Convergence

**Algorithm 2** Inference for E-RNN

**Require:** An instance $Z = [\mathbf{x}_1, \mathbf{x}_2, \ldots, \mathbf{x}_\omega]$, and a probability threshold $\hat{p}$.
1: **for** $t = 1, 2, \ldots, \omega$ **do**
2: $\quad p_t \leftarrow \text{MAX}(w^T o_t)$
3: $\quad \ell_{i,t} \leftarrow \text{ARGMAX}(w^T o_t)$
4: $\quad$ **if** $p_{i,t} \ge \hat{p}$ or $t = \omega$ **then**
5: $\quad\quad$ **return** $[\ell_{i,t}, p_{i,t}]$
6: $\quad$ **end if**
7: **end for**

We propose a thresholding based method that exploits the structure in label noise and is still simple enough to be implemented efficiently on top of LSTM training. Algorithm 1 presents a pseudo-code of our algorithm. The Train-LSTM procedure can use any standard optimization algorithm such as SGD+Momentum [31], Adam [22] to train a standard LSTM with a multi-class logistic regression layer. Input to this procedure is the instance level data constructed by windowing of the sequential data along with the corresponding estimated instance labels. Naturally, our technique applies even if we use any other architecture to learn the classifier $f$.

We further refine the trained LSTM by re-estimating the "correct" set of instances per sequential data point using a simple thresholding scheme. In particular, we find $k$ (a parameter) consecutive instances for which sum of predictions (using function $f$) for the data-point label $y$ is highest and include it in training set; the remaining set of instances are ignored for that particular iteration. The predicted label of a given point $X = [\mathbf{x}_1, \ldots, \mathbf{x}_T]$ is given by: $y = \arg\max_b \max_\tau \sum_{t \in [\tau, \tau+k-1]} f_b(\mathbf{x}_t)$.

While our algorithm is similar to the hard-thresholding approach popular in robust learning [6, 15], both the thresholding operator and the setting are significantly different. For example, robust learning techniques are expected to succeed when the amount of noisy labels is small, but in our problem only $k$ out of $T$ instances are correctly labeled (typically $k \ll T$). Nonetheless, we are still able to analyze our method in a simple setting where the data itself is sampled from a well-separated distribution.

### 3.1.1 Analysis

Let $Z = [(Z_1^P, \bar{\mathbf{y}}_1^P), \ldots, (Z_n^P, \bar{\mathbf{y}}_n^P), (Z_1^N, \bar{\mathbf{y}}_1^N), \ldots, (Z_n^N, \bar{\mathbf{y}}_n^N)]$ be the given dataset where $Z_i^P$ denotes the $i$-th "positive" bag and $Z_i^N$ denotes the $i$-th "negative" bag. For simplicity, we assume that each positive and negative bag contain $T$ points. $\bar{\mathbf{y}}_i^P$ denotes the label vector for $i$-th positive bag. For simplicity, we only consider binary classification problem in this section. That is, $\bar{\mathbf{y}}_i^P \in \{-1, 1\}^T$. Also, let number of positives (i.e. points with label $+1$) be $k$ and let all the positives be consecutive in index set $1 \le \tau \le T$. By definition, $\bar{\mathbf{y}}_{i,\tau}^N = -1$ for all instances in the bag. Note that $\bar{\mathbf{y}}_i^P$ are not available apriori. Let $S \subseteq [n] \times [T]$ be an index set of columns, i.e., $S = \{(i_1, \tau_1), \ldots, (i_{|S|}, \tau_{|S|})\}$.

Now, given $Z^P$, $Z^N$, the goal is to learn a linear classifier $\mathbf{w}$ s.t. $sign(\mathbf{w}^T Z_{i,\tau}^P) = \bar{\mathbf{y}}_{i,\tau}^P$ and $sign(\mathbf{w}^T Z_{i,\tau}^N) = -1$, i.e. each point in the positive/negative bag is correctly classified. To this end, we use the following modified version of Algorithm 1:

$$\mathbf{w}^0 = \arg\min_{\mathbf{w}} \sum_{i,\tau}(1 - \mathbf{w}^T Z_{i,\tau}^P)^2 + \sum_{i,\tau}(-1 - \mathbf{w}^T Z_{i,\tau}^N)^2,$$

$$S_{r+1} = \cup_{i=1}^n \cup_{\tau=k_1}^{k_1+k}(i, \tau), \text{where}, k_1 = \arg\max_{k_1} \sum_{\tau=k_1}^{k_1+k}\langle \mathbf{w}^r, Z_{i,\tau}^P \rangle, \forall i, \forall r \ge 0$$

$$\mathbf{w}^{r+1} = \arg\min_{\mathbf{w}} \sum_{(i,\tau) \in S_{r+1}}(1 - \langle \mathbf{w}, Z_{i,\tau}^P \rangle)^2 + \frac{1}{T}\sum_{i,\tau}(-1 - \langle \mathbf{w}, Z_{i,\tau}^N \rangle)^2, \quad \forall r \ge 0 \qquad (3.1.2)$$

where $k$ is a parameter that specifies the number of true positives in each bag, i.e., $\sum_\tau \mathbb{I}[y_{i,\tau}^P > 0] = k$.

Note that the above optimization algorithm is essentially an alternating optimization based algorithm where in each iteration we alternately estimate the classifier $\mathbf{w}$ and the *correct* set of positive points

in positive bags. Naturally the problem is non-convex and is in fact NP-hard in general, and hence the above algorithm might not even converge. However, by restricting the noise in the bags to a class of "nice" distributions, we can show that the above algorithm converges to the *global* optima at a linear rate. That is, within a small number of steps the algorithm is able to learn the *correct* set of positive points in each bag and hence the optimal classifier $\mathbf{w}^*$ as well.

**Theorem 3.1.** *Let* $Z^P = [Z_1^P, \ldots, Z_n^P]$, $Z^N = [Z_1^N, \ldots, Z_n^N]$ *be the data matrix of positive and negative bags, respectively, with each bag containing $T$ points, i.e.,* $Z_i^P = [\mathbf{x}_{i,1}^P, \ldots, \mathbf{x}_{i,T}^P]$ *and* $Z_i^N = [\mathbf{x}_{i,1}^N, \ldots, \mathbf{x}_{i,T}^N]$. *Let* $\bar{\mathbf{y}}^P$ *be the* **true** *labels of each data point in positive bags. Let each data point be given by:* $x_{i,\tau}^P = 0.5(\mathbf{y}_{i,\tau}^P + 1)\mu^+ + 0.5(1 - \mathbf{y}_{i,\tau}^P)\mu^- + \mathbf{g}_{i,\tau}^P$ *and* $x_{i,\tau}^N = \mu^- + \mathbf{g}_{i,\tau}^N$. *Let* $\mathbf{g}_{i,\tau}^N \overset{i.i.d}{\sim} \mathcal{D}$ *be sampled from $\mathcal{D}$ and let $\mathcal{D}$ be sub-Gaussian with sub-Gaussian norm $\psi(\mathcal{D})$. Wlog, let each $\mathbf{g}_{i,\tau}^N$ be isometric random vectors. Also, let $g_{i,\tau}^P$ be* **arbitrary** *vectors that satisfy:* $\|\sum_{(i,\tau)\in S} g_{(i,\tau)}^P\| \leq \gamma|S|$ *for all $S$ where $\gamma > 0$ is a constant. Moreover, let* $n \geq \frac{dTC_\psi^2}{k^2}$, *where $C_\psi^2 > 0$ is a constant dependent only on $\psi(\mathcal{D})$. Let* $\|\Delta_\mu\|^2 \geq 400C_\psi(\gamma^2+1) \cdot (\|\mu^+\| + \|\mu^-\|) \log(nT)$ *where $\Delta_\mu = \mu^+ - \mu^-$. Then,* $R = \log n$ *iterations of Algorithm in (3.1.2) recover the true positive set $S_* = \{(i,\tau), \bar{\mathbf{y}}_{i,\tau}^P = +1\}$* **exactly**, *with probability* $\geq 1 - 30/n^{20}$.

See supplement for a detailed proof of the above theorem.

**Remark 1**: Note that we allow positive set of points to be arbitrarily dependent on each other. The required condition on $G^P$ can be easily satisfied by dependent set of vectors, for example if $g_{i,\tau}^P \sim \frac{1}{d}g_{i,\tau-1}^P + N(0, I)$. Also, note that due to arbitrary dependence of "positive" set of points, the above problem is not a *homogeneous* MIML problem and has a more realistic model than extensively studied *homogeneous* MIML model [27]. Unfortunately, non-homogeneous MIML problems are in general NP-hard [27]. By exploiting structure in the problem, our thresholding based algorithm is still able to obtain the optimal solution with small computational complexity.

**Remark 2**: While our technique is similar to hard-thresholding based robust learning works [6, 15], our results hold despite the presence of large amounts of noise $(1 - k/T)$. Existing robust learning results require the fraction of errors to be less than a small constant and hence do not apply here.

**Remark 3**: While our current proof holds only for squared loss, we believe that similar techniques can be used for classification loss functions as well; further investigation is left for future work.

## 3.2 Early Classification (E-RNN)

LSTM models for classification tasks typically pipe the hidden state at the final time-step to a standard classifier like logistic regression to obtain the predicted label. As the number of time-steps $T$ can be large, going over all the time-steps can be slow and might exceed the prediction budget of tiny devices. Furthermore, in practice, most of the data points belong to "noise" class $(y = -1)$ which should be easy to predict early on. To this end, we teach the LSTM when it can stop early by piping the output of **each** step (instead of just the last step) of the LSTM to the classifier. The network is then trained to optimize the sum of loss of classifier's output at each step. That is,

$$\min \sum_i \sum_{t=1}^T \ell(w^T o_{i,t}) \tag{3.2.1}$$

where $w$ is the weight of the fully connected (F.C.) layer, and $o_{i,t}$ is the output of $t$-th step of the LSTM when processing data point $X_i$.

For inference, if at step $t$, a class is predicted with probability greater than a tunable threshold $\hat{p}$, we stop the LSTM and output the class. For efficiency, we predict the probability after every $\kappa$-steps. For wake-word detection type tasks where noise class forms an overwhelming majority of test points, we provide early classification only on the noise class (Figure 4c). Algorithm 2 describes our inference procedure for early classification. We overload notation and assume that $f(Z_{i,\tau})$ outputs a probability distribution over all classes.

## 3.3 EMI-RNN

MI-RNN and E-RNN are complementary techniques — MI-RNN reduces the number of time-steps in the sequential data point and E-RNN provides early prediction. Naturally, a combination of two

| Dataset | Hidden Dimension | LSTM | Accuracies for MI-RNN | | | Accuracies for E-RNN | | | |
|---|---|---|---|---|---|---|---|---|---|
| | | | MI-RNN at Round 0 | MI-RNN | Computation saved in % | Early prediction at 55% time steps | | Early prediction at 75% time steps | |
| | | | | | | % Predicted early | Overall accuracy | % predicted early | Overall accuracy |
| HAR-6 | 8 | 89.54 | 90.83 | 91.92 | 62.5 | 79.74 | 89.75 | 81.50 | 89.78 |
| | 16 | 92.90 | 92.16 | 93.89 | | 86.80 | 91.24 | 87.68 | 91.24 |
| | 32 | 93.04 | 93.75 | 91.78 | | 85.06 | 91.75 | 85.88 | 91.85 |
| Google-13 | 16 | 86.99 | 88.06 | 89.78 | 50.5 | 35.08 | 84.18 | 55.14 | 84.27 |
| | 32 | 89.84 | 91.80 | 92.61 | | 41.05 | 88.31 | 64.41 | 88.42 |
| | 64 | 91.13 | 92.87 | 93.16 | | 59.13 | 92.24 | 85.74 | 92.43 |
| STCI-2 | 8 | 98.07 | 97.38 | 98.08 | 50 | 50.17 | 96.39 | 74.94 | 96.52 |
| | 16 | 98.78 | 98.35 | 99.07 | | 54.15 | 98.16 | 84.53 | 98.35 |
| | 32 | 99.01 | 98.50 | 98.96 | | 53.25 | 98.24 | 81.89 | 98.32 |
| GesturePod-6 | 16 | - | 96.23 | 98.00 | 50 | – | – | – | – |
| | 32 | 94.04 | 98.27 | 99.13 | | 39.38 | 84.48 | 58.93 | 84.48 |
| | 48 | 97.13 | 98.27 | 98.43 | | 76.68 | 96.39 | 99.13 | 96.55 |
| DSA-19 | 32 | 84.56 | 86.97 | 87.01 | 28 | 55.48 | 83.72 | 56.40 | 83.68 |
| | 48 | 85.35 | 84.42 | 89.60 | | 68.81 | 82.63 | 69.16 | 82.54 |
| | 64 | 85.17 | 85.08 | 88.11 | | 41.00 | 85.48 | 41.27 | 85.52 |

Table 1: Accuracies of MI-RNN and E-RNN methods compared to a baseline LSTM on the five datasets. Each row corresponds to experiments with a fixed hidden dimension. The first column under MI-RNN (at Round 0) lists the test accuracy of an LSTM trained on instances (windows) before any refinement of instance labels. The next column lists the test accuracy after the completion of MI-RNN. The Third column lists the computation saved due to the fewer number of steps needed for the shorter windows. The columns under E-RNN list the fraction examples that the E-RNN can predict early at the 55% (and 75%) time step and the overall prediction accuracy.

techniques should provide an even more efficient solution. Interestingly, here the total benefit is larger than the sum of parts. Early classification should become more effective once MI-RNN identifies instances that contain true and tight class signatures as these signatures are unique to that class. However, since the two methods are trained separately, a naïve combination ends up reducing the accuracy significantly in many cases (see Figure 4).

To alleviate this concern, we propose an architecture (see Figure 2) that trains MI-RNN and E-RNN jointly. That is, we break each data point into multiple instances as in 3.1.1 and replace the loss function in the Train-LSTM procedure in Algorithm 1 with the sum loss function in equation 3.2.1. We call this modified version of Algorithm 1 Early Multi-Instance RNN (EMI-RNN). EMI-RNN is able to learn an LSTM that is capable of predicting short signatures accurately as well as stopping early. For instance, on GesturePod-6 dataset early classification at 55% time-step leads to a 10% drop in accuracy for E-RNN. With EMI-RNN, not only is this accuracy drop recovered but an additional 3% improvement and a computational saving of saving 80% is obtained (see Table 1, Figure 3).

For training, we first apply Algorithm 1 to train MI-RNN and form a new training data $D = \{Z_{i,\tau}, \tau \in [s_i, s_i + k - 1], \forall 1 \leq i \leq n\}$ by pruning each bag of instances $\mathcal{X}_i$ down to the set of *positive* instances $Z_{i,\tau}$. We then use $D$ to train LSTM with the E-RNN loss function (3.2.1) for ensuring accurate prediction with early classification. For inference, we use the tunable Algorithm 2 for early prediction for each instance and then compute $y = \arg\max_b \max_\tau \sum_{t \in [\tau, \tau+k-1]} f_b^{E-RNN}(\mathbf{x}_t)$, where $f^{E-RNN}$ is computed using our jointly trained LSTM model $f$ applied to Algorithm 2.

## 4 Empirical Results

We empirically evaluate our methods on five sequential (time-series) datasets – three activity/gesture recognition datasets and two audio related datasets. The details of the datasets including their sources are reported in Table 4 in the supplement. HAR-6, GesturePod-6 and DSA-19 are multi-class activity recognition datasets. Of the three, only GesturePod-6 data set has negative examples. Google-

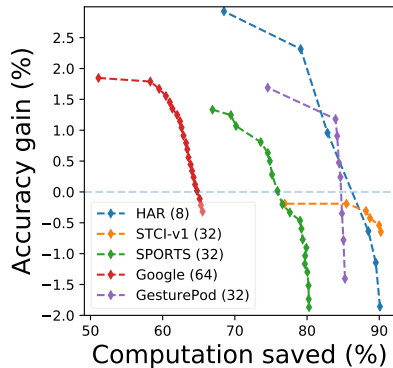

Figure 3: Trade-off between accuracy gains and computational savings of EMI-RNN over baseline method. Hidden dimension listed in parenthesis.

| Device | Hidden Dim. | LSTM | MI-RNN | EMI-RNN |
|---|---|---|---|---|
| RPi0 (22.50 ms) | 16 | 28.14 | 14.06 | 5.62 |
| | 32 | 74.46 | 37.41 | 14.96 |
| | 64 | 226.1 | 112.6 | 45.03 |
| RPi3 (26.39 ms) | 16 | 12.76 | 6.48 | 2.59 |
| | 32 | 33.10 | 16.47 | 6.58 |
| | 64 | 92.09 | 46.28 | 18.51 |

Table 2: Prediction time in milliseconds for keyword spotting (Google-13) on Raspberry Pi 0 and 3 for a simple LSTM implementation in C. Constraints for real-time detection are listed in brackets.

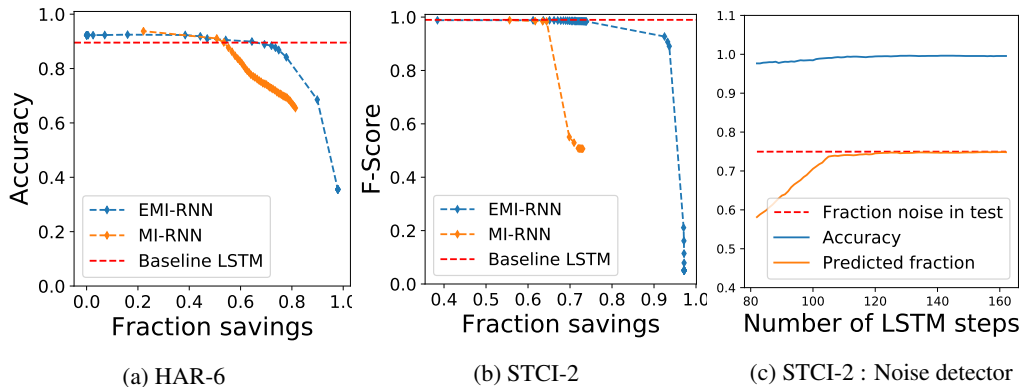

(a) HAR-6        (b) STCI-2        (c) STCI-2 : Noise detector

Figure 4: Here, a and b compare the accuracy and computational saving of early classification (Algorithm 2) on EMI-RNN model vs MI-RNN model. Clearly, on a fixed computational savings requirement, joint training of EMI-RNN provides significantly more accurate models. In c, early noise detection with E-RNN in STCI-2is shown. It can be seen that the drop in accuracy on the predicted fraction is very low (hidden dimension 32).

13 dataset is a multi-class keyword detection dataset where all but 12 classes are treated as negative examples. STCI-2 is a proprietary audio dataset where the goal is to recognize a wake-word (e.g., Hey Cortana) from the background. Although not a prerequisite, in these datasets, the actual signature of the activity or audio example is shorter that the length of the training examples.

For training, in addition to the RNN's hyperparameters, EMI-RNN requires the selection of three more hyperparameters. There is a principled way of selecting them. The instance width $\omega$ can be set from domain knowledge of the signature (e.g. longest length of keyword in keyword spotting), or can be found by cross-validation (Figure 5a in supplement). The stride between instances is set to the value typically used for vanilla RNN based on domain knowledge (e.g. ~25ms for audio) or based on cross-validation. The value of $k$, the number of consecutive instances that contain a signature, is set to $\lceil \omega / \text{instance stride} \rceil$.

We use the standard accuracy measure to compare various methods. That is, for a given algorithm, we compute predicted label $\widehat{y}_j$ for each sequential data point $X_j^{\text{test}}$, $1 \leq j \leq m$ in the test set and then compute accuracy as $\frac{1}{m} \sum_{i=1}^{m} \mathbf{I}[\widehat{y}_i == y_i]$. Table 1 compares the accuracies of the MI-RNN and E-RNN methods with that of an LSTM trained on the full-length training data $[(X_1, y_1), \ldots, (X_n, y_n)]$. The results in Table 1 show that for a fixed hidden dimension, MI-RNN can save the computation needed for inference by up to a factor of two, and simultaneously improve the prediction accuracy. Accuracy gain of MI-RNN over baseline is significantly higher for small model sizes, e.g., gain of 2.5% for Google-13, as MI-RNN is able to prune away noisy instances. Further, it also demonstrates

| Dataset | LSTM | | EMI-RNN | | Speedup | Speedup @~1% Drop |
|---------|------|------|---------|------|---------|------|
| | Hidden Dim. | Accuracy | Hidden Dim | Accuracy | | |
| HAR-6 | 32 | 93.04 | 16 | 93.89 | 10.5x | 42x |
| Google-13 | 64 | 91.13 | 32 | 92.61 | 8x | 32x |
| STCI-2 | 32 | 99.01 | 16 | 99.07 | 8x | 32x |
| GesturePod-6 | 48 | 97.13 | 8 | 98.00 | 72x | - |
| DSA-19 | 64 | 85.17 | 32 | 87.01 | 5.5x | - |

Table 3: The table shows the hidden-size of EMI-RNN required to achieve same or slightly higher accuracy than baseline LSTM, and the corresponding inference speed-up over LSTM. Last column shows inference cost speed-up provided by EMI-RNN if the model-size is decreased further while ensuring that the accuracy is at most 1% lower than LSTM's accuracy.

that E-RNN can train a model that can accurately predict a large fraction of time-series data points early. At 55% time steps, we notice that up to 80% of the data points can be confidently predicted. In real world tasks such as wake-word detection, most data points do not contain positive signatures and are hence juse noise. In such situations, it is critical to eliminate the negative examples (i.e., noise) as early as possible. Figure 4c demonstrates that around the 100-th time step (of 160), noise can be identified with ~100% accuracy.

The drawback of E-RNN is that on some datasets, the overall accuracy, which includes data points that could not be classified until the last step, degrades. In one instance (GesturePod-6, 32 dim.), accuracy drops almost 10%. The problem can be addressed by using the EMI-RNN method that jointly trains for the right window labels and for early classification. Figure 3 demonstrates the trade-off between accuracy gain of the LSTM trained by the EMI-RNN method (over the baseline LSTM) and the percentage of computation saved due to fewer LSTM steps, at a fixed hidden dimension. The trade-off is tuned by adjusting the probability threshold $\hat{p}$ for early prediction. In all cases except STCI-2, EMI-RNN can save $65\% - 90\%$ of the computation without losing accuracy compared to the baseline. In fact, in most cases EMI-RNN can provide up to 2% more accurate predictions while providing $2 - 3\times$ saving in computation. Similarly, we can trade-off accuracy for higher computation speed-up, i.e., EMI-RNN outperforms baseline LSTM models using less than half the number of states thus providing huge computational gains (Table 3).

Improvements in computation time can critically enable real-time predictions on tiny devices such as Raspberry Pi. Consider the example of spotting keyword as in Google-13. Typically, predictions on audio samples are made on sliding windows of 30ms, i.e., every 30ms, we extract spectral features and start a new LSTM to predict the keyword in the trailing window. For real-time prediction, this must be completed within 30ms. Table 2 demonstrates that while normal LSTM cannot meet this deadline, LSTMs trained with MI-RNN and EMI-RNN techniques comfortably accommodate a 32 hidden dimensional LSTM while learning a model that is 1.5% more accurate. Note that the times for EMI-RNN in the table were estimated by choosing the probability threshold $\hat{p}$ that provides at least 1% accuracy improvement over the baseline.

## 5 Conclusion

This paper proposed EMI-RNN algorithm for sequential data classification. EMI-RNN was based on a multi-instance learning (MIL) formulation of the sequential data classification problem and exploited techniques from robust learning to ensure efficient training. Analysis of EMI-RNN showed that it can be efficiently trained to recover globally optimal solution in interesting non-homogeneous MIL settings. Furthermore, on several benchmarks, EMI-RNN outperformed baseline LSTM while providing up to 70x reduction in inference cost.

While this paper restricted it's attention to fixed length sensors related problems, application of similar approach to the natural language processing problems should be of interest. Relaxation of restrictive assumptions on data distributions for analysis of EMI-RNN should also be of wide interest.

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
