[Supplementary Material]

# A  Proof of Theorem 3.1

## A.1  Preliminaries

In this section, we introduce certain key tools and lemmas that we need for our proof.

**Definition A.1.** $a \sim \mathcal{D}$ *is defined to be a sub-Gaussian isotropic random vector with sub-Gaussian norm* $\psi(D)$ *if:*

$$\mathbb{E}\left[\mathbf{a}\right] = 0, \mathbb{E}\left[\mathbf{a}\mathbf{a}^T\right] = I, \psi(D) = \sup_{\mathbf{v} \in S^{d-1}} \sup_{p \geq 1} p^{-\frac{1}{2}} \left(\mathbb{E}_{\mathbf{a} \sim \mathcal{D}}|\langle \mathbf{a}, \mathbf{v}\rangle|^p\right)^{1/p}$$

**Theorem A.2** (Theorem 5.39 of [34]). *Let A be an* $d \times n$ *matrix who columns* $A_i$ *are independent sub-Gaussian isotropic random vectors in* $\mathbb{R}^d$ *sampled i.i.d. from distribution* $\mathcal{D}$. *Then, the following holds w.p.* $\geq 1 - \delta$:

$$\sqrt{n} - C_\psi \sqrt{d} - \sqrt{c_\psi \log(1/\delta)} \leq s_{min}(A) \leq s_{max}(A) \leq \sqrt{n} + C_\psi \sqrt{d} + \sqrt{c_\psi \log(1/\delta)},$$

*where* $s_{min}$ *is the smallest singular value of A,* $s_{max}$ *is the largest singular value of A.* $C_\psi > 0, c_\psi > 0$ *are constants dependent only on the sub-Gaussian norm* $\psi$ *of* $\mathcal{D}$ *(see Definition A.1).*

We also need the following lemma about sub-Gaussian rows.

**Lemma A.3.** *Let A be an* $d \times n$ *matrix who columns* $A_i \overset{i.i.d.}{\sim} \mathcal{D} \in \mathbb{R}^d$ *and let* $S \in [n]$ *be a* **fixed** *index set. Let* $A_S \in \mathbb{R}^{d \times |S|}$ *contains columns of A in index set S. Then, the following holds w.p.* $\geq 1 - \delta$:

$$\|A_S \mathbf{1}_S\| \leq \sqrt{|S|} \left(\sqrt{d} + C_\psi \sqrt{\log \frac{1}{\delta}}\right),$$

*where* $C_\psi > 0$ *is a constant dependent only on the sub-Gaussian norm* $\psi$ *of* $\mathcal{D}$.

**Lemma A.4.** *Let* $E \in \mathbb{R}^{d \times d}$ *be such that* $\|E\|_2 \leq 1/2$, *then the following holds for all* $\mathbf{a}, \mathbf{b} \in \mathbb{R}^d$:

$$\mathbf{a}^T (I + E)^{-1} \mathbf{b} \leq \mathbf{a}^T \mathbf{b} + \|E\|_2 \cdot \|\mathbf{a}\|\|\mathbf{b}\|.$$

**Lemma A.5.** *Let* $M \in \mathbb{R}^{d \times 2}$ *and let* $C_r = MM^T + \Sigma_r$. *Let* $s_{min}(M^T \Sigma_r^{-1} M) > 0$. *Then for all* $\mathbf{v} \in \mathbb{R}^2$, *we have:*

$$\frac{s_{min}(M^T \Sigma_r^{-1} M)}{1 + s_{min}(M^T \Sigma_r^{-1} M)} \|\mathbf{v}\|^2 \leq \mathbf{v}^T M^T C_r^{-1} M \mathbf{v}.$$

*Furthermore, if* $s_{min}(M^T \Sigma_r^{-1} M) \geq 2$, *we have:*

$$\frac{2}{3} \cdot \|\mathbf{v}\|^2 \leq \mathbf{v}^T M^T C_r^{-1} M \mathbf{v}.$$

*Finally, if* $s_{min}(M^T \Sigma_r^{-1} M) \geq 2$, *then the following holds for all* $u, v$:

$$\mathbf{u}^T C_r^{-1} M \mathbf{v} = \|(M^T \Sigma_r^{-1} M)^{-1}\| \cdot \|\mathbf{u}^T \Sigma_r^{-1} M\| \cdot \|\mathbf{v}\|$$

*Proof.* Using Sherman-Morrison-Woodbury formula:

$$C_r^{-1} = \Sigma_r^{-1} - \Sigma_r^{-1} M (I + M^T \Sigma_r^{-1} M)^{-1} M^T \Sigma_r^{-1}.$$

That is,

$$C_r^{-1} M \mathbf{v} = \Sigma_r^{-1} M (I - (I + M^T \Sigma_r^{-1} M)^{-1} M^T \Sigma_r^{-1} M) \mathbf{v} = \Sigma_r^{-1} M (I + M^T \Sigma_r^{-1} M)^{-1} \mathbf{v}. \tag{A.1.1}$$

Hence,

$$\mathbf{v}^T M C_r^{-1} M \mathbf{v} = \mathbf{v}^T A \mathbf{v} - \mathbf{v}^T A (I + A)^{-1} A \mathbf{v} = \mathbf{v}^T (I + A^{-1})^{-1} \mathbf{v},$$

where $A = M^T \Sigma_r^{-1} M$. First part of Lemma now follows by using the assumption that $s_{min}(A) \geq 2$. Similarly,

$$\mathbf{u}^T C_r^{-1} M \mathbf{v} = \mathbf{u}^T \Sigma_r^{-1} M \mathbf{v} - \mathbf{u}^T \Sigma_r^{-1} M (I + (M^T \Sigma_r^{-1} M)^{-1})^{-1} \mathbf{v}$$

$$\leq \|(M^T \Sigma_r^{-1} M)^{-1}\|\|\mathbf{u}^T \Sigma_r^{-1} M\|\|\mathbf{v}\|. \tag{A.1.2}$$

$\square$

## A.2 Main Proof

We first introduce some notation for our proof. Let, $\mathbf{x}_{i,\tau}^P = 0.5(\mathbf{y}_{i,\tau}^P+1)\mu^+ +0.5(1-\mathbf{y}_{i,\tau}^P)\mu^- +\mathbf{g}_{i,\tau}^P$ be the $i$-th positive bag's $\tau$-th data point. Similarly, let $x_{i,\tau}^N = \mu^- + \mathbf{g}_{i,\tau}^N$ be the $i$-th negative bag's $\tau$-th data point. $Z^P \in \mathbb{R}^{d\times nT}$ denotes the data matrix for positive bags, where $((i-1)\cdot T+\tau)$-th column of $Z^P$ is given by $Z_{(i,\tau)}^P = \mathbf{x}_{i,\tau}^P$. Similarly, $G^P \in \mathbb{R}^{d\times nT}$ and $G^N \in \mathbb{R}^{d\times nT}$ contain the noise term in each point s.t. $((i-1)\cdot T+\tau)$-th column of $G^P$ is $G_{(i,\tau)}^P = \mathbf{g}_{i,\tau}$ and $((i-1)\cdot T+\tau)$-th column of $G^N \in \mathbb{R}^{d\times nT}$ is given by: $G_{(i,\tau)}^N = \mathbf{g}_{i,\tau}^N$. That is, $G_S^P \in \mathbb{R}^{d\times|S|}$ s.t. $\ell$-th column of $G_S^P$ is given by $G_{(i_\ell,\tau_\ell)}^P$.

*Proof of Theorem 3.1.* Note that set $S_r$ is an estimate for the true set of positives i.e., $S^* = \{(i,\tau), \bar{y}_{i,\tau}^P = +1\}$. Also let $\beta_r \cdot n = |S_r\backslash S_*|$ be the number of "incorrect" elements in $S_r$. Now, our proof relies on two key results: a) assuming $\beta_r$ is small, we show that $\beta_{r+1}\cdot n = |S_{r+1}\backslash S_*|$ decreases by a constant multiplicative factor, b) our initial estimate $\mathbf{w}^0$ ensures that $\beta_1 \cdot n = |S_1\backslash S_*|$ is indeed small, hence we can apply the result in (a) inductively to obtain the result. In particular, using Theorem A.7 and Lemma A.6, we have after $T$ round:

$$\beta_{r+1} \leq .99^{-R}.$$

Hence, after $R = \log n$ steps, we have $S_{r+1} = S_*$. $\qquad\square$

**Lemma A.6.** *Consider the setting of Theorem 3.1. Also, let $S_1$ be computed using (3.1.2). Let $\Delta_\mu = \mu^+ - \mu^-$ and $\|\Delta_\mu\| \geq 400C_\psi\gamma \cdot \log(nT)$. Then, the following holds w.p. $\geq 1 - 30/n^{20}$:*

$$\beta_1 \leq \frac{1}{20\gamma \cdot C_\psi \cdot \sqrt{\log(nT)}}.$$

*Proof.* As $\mathbf{w}^0$ is the least squares solution to first equation of (3.1.2), we have:

$$\mathbf{w}^0 = C_0^{-1}(Z^P\mathbf{1} - Z^N\mathbf{1}),$$

where,

$$C_0 = nk\cdot\mu^+(\mu^+)^T + n(2T-k)\cdot\mu^-(\mu^-)^T + G^P(G^P)^T + G^N(G^N)^T$$
$$= nk\cdot\mu^+(\mu^+)^T + n(2T-k)\cdot\mu^-(\mu^-)^T + \Sigma_0, \quad \text{(A.2.1)}$$

where $\Sigma_0 = G^P(G^P)^T + G^N(G^N)^T$. Using Theorem A.2, we have (w.p. $\geq 1 - 1/n^{20}$):

$$\Sigma_0 \succeq 2nT(I+E), \text{ where, } \|E\|_2 \leq C_\psi\sqrt{\frac{d+\log n}{2nT}} \leq \frac{1}{10}, \quad \text{(A.2.2)}$$

where $C_\psi$ is a constant dependent only on $\psi(\mathcal{D})$ and $nT \geq 100C_\psi^2(d+\log n)$.

Also, using (3.1.2), we have:

$$\mathbf{1}^T(Z_{S_1}^P - Z_{S_*}^P)^T C_0^{-1}(Z^P\mathbf{1} - Z^N\mathbf{1}) \geq 0,$$
$$i.e., \mathbf{1}^T(Z_{S_*\backslash S_1}^P - Z_{S_1\backslash S_*}^P)^T C_0^{-1}(Z^P\mathbf{1} - Z^N\mathbf{1}) \leq 0. \quad \text{(A.2.3)}$$

Note that $Z_{S_*\backslash S_1}^P\mathbf{1} = \beta_1 nk\cdot\mu^+ + G_{S_*\backslash S_1}^P\mathbf{1}$ and $Z_{S_1\backslash S_*}^P\mathbf{1} = \beta_1 nk\cdot\mu^- + G_{S_1\backslash S_*}^P\mathbf{1}$. Similarly, $Z^P\mathbf{1} - Z^N\mathbf{1} = nk(\mu^+ - \mu^-) + G^P\mathbf{1} - G^N\mathbf{1}$. Combining these observations with (A.2.3), we have:

$$Q_1^2 + Q_2 + Q_3 + Q_4 \leq 0,$$
$$Q_1^2 = \beta_1 n^2 k^2 \cdot \Delta_\mu^T C_0^{-1}\Delta_\mu, \qquad Q_2 = nk\cdot\mathbf{b}_1^T C_0^{-1}\Delta_\mu,$$
$$Q_3 = \beta_1 nk\cdot\mathbf{a}_0^T C_0^{-1}\Delta_\mu, \qquad Q_4 = \mathbf{b}_1 C_0^{-1}\mathbf{a}_0, \quad \text{(A.2.4)}$$

where $\mathbf{b}_1 := (G_{S_*\backslash S_1}^P - G_{S_1\backslash S_*}^P)\mathbf{1}$ and $\mathbf{a}_0 = (G^P - G^N)\mathbf{1}$. Now, using Lemma A.5 and (A.2.2), we have:

$$\frac{1}{nk}\cdot\frac{k\|\mu^+ - \mu^-\|^2}{4T + k\|\mu^+ - \mu^-\|^2} \leq \Delta_\mu^T C_0^{-1}\Delta_\mu, \qquad \beta_1 nk\cdot\frac{k\|\mu^+ - \mu^-\|^2}{4T + k\|\mu^+ - \mu^-\|^2} \leq Q_1^2, \quad \text{(A.2.5)}$$

where we use the fact that $s_{min}([\mu^+\ \mu^-])^2 \geq .5\|\mu^+ - \mu^-\|^2$.

Now, we consider $Q_2$:

$$\frac{1}{nk}Q_2 = \frac{\mathbf{b}_1^T\Delta_\mu}{\|\Delta_\mu\|^2}\Delta_\mu^T C_0^{-1}\Delta_\mu + \mathbf{b}_\perp^T C_0 \Delta_\mu \leq \frac{\|\mathbf{b}_1\|}{\|\Delta_\mu\|}\Delta_\mu^T C_0^{-1}\Delta_\mu + \mathbf{b}_\perp^T C_0^{-1}\Delta_\mu, \quad \text{(A.2.6)}$$

where $\mathbf{b}_\perp = (I - \Delta_\mu\Delta_\mu^T/\|\Delta_\mu\|^2)\mathbf{b}_1$.

We need to consider two cases now: a) if $\|\Delta_\mu\|^2 \leq 10T/k$, and b) if $\|\Delta_\mu\|^2 \geq 10T/k$. Proof for the second case follows using similar arguments to Theorem A.7. So, here we focus on the case when $\|\Delta_\mu\|^2 \leq 10T/k$. Let $M = [\sqrt{nk}\mu^+\ \sqrt{2nT - nk}\mu^-]$. Then, using Sherman-Morrison-Woodbury formula, we have:

$$\mathbf{b}_\perp^T C_0^{-1}\Delta_\mu = \mathbf{b}_\perp^T\Sigma_0^{-1}\Delta_\mu - \mathbf{b}_\perp^T\Sigma_0^{-1}M(I + M^T\Sigma_0^{-1}M)^{-1}M^T\Sigma_0^{-1}\Delta_\mu,$$

$$|\mathbf{b}_\perp^T C_0^{-1}\Delta_\mu| \leq \frac{1.2}{nT}\|\mathbf{b}_1\|\|\Delta_\mu\|, \quad \text{(A.2.7)}$$

where the inequality follows using $\|\mathbf{b}_\perp\| \leq \|\mathbf{b}_1\|$, $\|\Sigma_0 - 2nT \cdot I\| \leq \|E\| \leq \frac{1}{10}$, and $\|\Sigma_0^{-1/2}M(I + M^T\Sigma_0^{-1}M)^{-1}M^T\Sigma_0^{-1/2}\|_2 \leq 1$. The above inequality holds with probability $\geq 1 - 10/n^{20}$. Using (A.2.6), (A.2.7), we have (w.p. $\geq 1 - 20/n^{20}$):

$$|Q_2| \leq nk\|\mathbf{b}_1\|\left(\frac{2}{\|\Delta_\mu\|}\Delta_\mu^T C_0^{-1}\Delta_\mu + \frac{2}{nT}\cdot\|\Delta_\mu\|\right). \quad \text{(A.2.8)}$$

Using same argument as above, the following holds (w.p. $\geq 1 - 21/n^{20}$):

$$|Q_3| \leq \beta_1 nk \cdot \|\mathbf{a}_0\| \cdot \left(\frac{2}{\|\Delta_\mu\|}\Delta_\mu^T C_0^{-1}\Delta_\mu + \frac{2}{nT}\cdot\|\Delta_\mu\|\right). \quad \text{(A.2.9)}$$

Finally, the following holds (w.p. $\geq 1 - 21/n^{20}$):

$$|Q_4| \leq \frac{\|\mathbf{b}_1\|}{nT}\cdot\|\mathbf{a}_0\|. \quad \text{(A.2.10)}$$

Now, using Lemma A.3 and the assumption on $G_{S*}^P$, we have (w.p. $\geq 1 - 5/n^{20}$):

$$\|\mathbf{b}_1\| \leq \sqrt{\beta_1 nk}(\sqrt{d} + C_\psi\sqrt{\beta_1 nk\log nT}) + \gamma\beta_1 nk, \|\mathbf{a}_0\| \leq \sqrt{nk}(\sqrt{d} + C_\psi\sqrt{\log nT}) + \gamma nk, \quad \text{(A.2.11)}$$

Combining (A.2.4), (A.2.8), (A.2.9), (A.2.10), (A.2.11), we have (w.p. $\geq 1 - 25/n^{20}$):

$$\left(\beta_1 n^2 k^2 - \frac{2(\|\mathbf{b}_1\| + \beta_1\|\mathbf{a}_0\|)nk}{\|\Delta_\mu\|}\right)\cdot\Delta_\mu^T C_0^{-1}\Delta_\mu - \frac{2nk(\|\mathbf{b}_1\| + \beta_1\|\mathbf{a}_0\|)}{nT}\|\Delta_\mu\| - \frac{\|\mathbf{b}_1\|\|\mathbf{a}_0\|}{nT} \leq 0 \quad \text{(A.2.12)}$$

Using (A.2.9), (A.2.11), and the assumption that $\|\Delta_\mu\| \geq 20C_\psi\gamma \cdot \log(nT)$ and $n \geq \frac{d \cdot T \cdot C_\psi^2}{k^2}$, we note that coefficient of $\Delta_\mu^T C_0^{-1}\Delta_\mu$ term above is positive and greater than $\beta_1 n^2 k^2/2$. Lemma now follows by combining (A.2.9), (A.2.11) with above equation. $\square$

**Theorem A.7.** *Consider the setting of Theorem 3.1. Also, let $S_r, S_{r+1}$ be computed using (3.1.2) in $t$-th and $(t+1)$-th iteration, respectively. Let $\Delta_\mu = \mu^+ - \mu^-$, $\|\Delta_\mu\|^2 \geq 400C_\psi \cdot (\|\mu^+\| + \|\mu^-\|)\log(nT)$. Then, the following holds w.p. $\geq 1 - 30/n^{20}$:*

$$\beta_{r+1} \leq 0.9\beta_r.$$

*Proof.* As $\mathbf{w}^r$ is the least squares solution to third equation of (3.1.2), we have:

$$\mathbf{w}^r = C_r^{-1}(Z_{S_r}^P\mathbf{1} - \frac{k}{T}Z^N\mathbf{1}),$$

where,

$$C_r = (1 - \beta_r)nk \cdot \mu^+(\mu^+)^T + (1 + \beta_r)nk \cdot \mu^-(\mu^-)^T + G_{S_r}^P(G_{S_r}^P)^T + \frac{k}{T}G^N(G^N)^T$$

$$= (1 - \beta_r)nk \cdot \mu^+(\mu^+)^T + (1 + \beta_r)nk \cdot \mu^-(\mu^-)^T + \Sigma_r, \quad \text{(A.2.13)}$$

where $\Sigma_r = G_{S_r}^P (G_{S_r}^P)^T + \frac{k}{T} G^N (G^N)^T$. Using Theorem A.2, we have (w.p. $\geq 1 - 1/n^{20}$):

$$\Sigma_r = nk(I + E) + G_{S_r}^P (G_{S_r}^P)^T, \text{ where, } \|E\|_2 \leq C_\psi \sqrt{\frac{d + \log n}{nT}} \leq \frac{1}{10}, \qquad (A.2.14)$$

where $C_\psi$ is a constant dependent only on $\psi(\mathcal{D})$ and $nT \geq 100 C_\psi^2 (d + \log n)$.

Also, using (3.1.2), we have:

$$\mathbf{1}^T (Z_{S_{r+1}}^P - Z_{S_*}^P)^T C_r^{-1} (Z_{S_r}^P \mathbf{1} - \frac{k}{T} Z^N \mathbf{1}) \geq 0,$$

$$i.e., \mathbf{1}^T (Z_{S_* \setminus S_{r+1}}^P - Z_{S_{r+1} \setminus S_*}^P)^T C_r^{-1} (Z_{S_r}^P \mathbf{1} - \frac{k}{T} Z^N \mathbf{1}) \leq 0. \qquad (A.2.15)$$

Note that $Z_{S_* \setminus S_r}^P \mathbf{1} = \beta_{r+1} nk \cdot \mu^+ + G_{S_* \setminus S_r}^P$ and $Z_{S_r \setminus S_*}^P \mathbf{1} = \beta_{r+1} nk \cdot \mu^- + G_{S_{r+1} \setminus S_*}^P$. Similarly, $Z_{S_r}^P \mathbf{1} - \frac{k}{T} Z^N \mathbf{1} = (1 - \beta_r) nk (\mu^+ - \mu^-) + G_{S_r}^P \mathbf{1} - \frac{k}{T} G^N \mathbf{1}$. Combining these observations with (A.2.15), we have:

$$Q_1^2 + Q_2 + Q_3 + Q_4 \leq 0,$$
$$Q_1^2 = \beta_{r+1}(1 - \beta_r) n^2 k^2 \cdot \Delta_\mu^T C_r^{-1} \Delta_\mu, \qquad Q_2 = (1 - \beta_r) nk \cdot \mathbf{b}_{r+1}^T C_r^{-1} \Delta_\mu,$$
$$Q_3 = \beta_{r+1} nk \cdot \mathbf{a}_r^T C_r^{-1} \Delta_\mu, \qquad Q_4 = \mathbf{b}_{r+1} C_r^{-1} \mathbf{a}_r, \qquad (A.2.16)$$

where $\mathbf{b}_{r+1} := (G_{S_* \setminus S_{r+1}}^P - G_{S_{r+1} \setminus S_*}^P) \mathbf{1}$ and $\mathbf{a}_r = (G_{S_r}^P - \frac{k}{T} G^N) \mathbf{1}$. Now, using Lemma A.5 and (A.2.14), we have:

$$\frac{1}{nk} \cdot \frac{2\|\mu^+ - \mu^-\|^2}{2 + (1 - \beta_r)\|\mu^+ - \mu^-\|^2} \leq \Delta_\mu^T C_r^{-1} \Delta_\mu, \ \beta_{r+1} nk \cdot \frac{2\|\mu^+ - \mu^-\|^2}{2 + (1 - \beta_r)\|\mu^+ - \mu^-\|^2} \leq Q_1^2, \qquad (A.2.17)$$

where we use the fact that $s_{min}([\mu^+ \ \mu^-])^2 \geq .5\|\mu^+ - \mu^-\|^2$.

Now, we consider $Q_2$:

$$\frac{1}{(1 - \beta_r) nk} Q_2 = \frac{\mathbf{b}_{r+1}^T \Delta_\mu}{\|\Delta_\mu\|^2} \Delta_\mu^T C_r^{-1} \Delta_\mu + \mathbf{b}_\perp^T C_r \Delta_\mu \leq \frac{\|\mathbf{b}_{r+1}\|}{\|\Delta_\mu\|} \Delta_\mu^T C_r^{-1} \Delta_\mu + \mathbf{b}_\perp^T C_r^{-1} \Delta_\mu, \qquad (A.2.18)$$

where $\mathbf{b}_\perp = (I - \Delta_\mu \Delta_\mu^T / \|\Delta_\mu\|^2) \mathbf{b}_{r+1}$.

We consider the second term above. Let $M = \sqrt{(1 - \beta_r) nk} [\mu^+ \ \mu^-]$. Then, using Lemma A.5 we have (w.p. $\geq 1 - 20/n^{20}$):

$$|\mathbf{b}_\perp^T C_r^{-1} \Delta_\mu| \leq \|\mathbf{b}_{r+1}\| \cdot \frac{4}{\sqrt{1 - \beta_r} \|\Delta_\mu\|^2} \cdot \frac{\|\mu^+\| + \|\mu^-\|}{nk}, \qquad (A.2.19)$$

where the inequality follows using $\|\mathbf{b}_\perp\| \leq \|\mathbf{b}_{r+1}\|$, $\|(M^T \Sigma_r^{-1} M)^{-1}\| \leq \frac{2}{(1 - \beta_r)\|\Delta_\mu\|^2}$, and $\|M\| \leq \sqrt{nk}(\|\mu^+\| + \|\mu^-\|)$. Using (A.2.18), (A.2.19), we have (w.p. $\geq 1 - 20/n^{20}$):

$$|Q_2| \leq (1 - \beta_r) nk \|\mathbf{b}_{r+1}\| \left( \frac{1}{\|\Delta_\mu\|} \Delta_\mu^T C_r^{-1} \Delta_\mu + \frac{4}{\sqrt{1 - \beta_r} \|\Delta_\mu\|^2} \cdot \frac{\|\mu^+\| + \|\mu^-\|}{nk} \right). \quad (A.2.20)$$

Using same argument as above, the following holds (w.p. $\geq 1 - 21/n^{20}$):

$$|Q_3| \leq \beta_{r+1} nk \cdot \|\mathbf{a}_r\| \cdot \left( \frac{1}{\|\Delta_\mu\|} \Delta_\mu^T C_r^{-1} \Delta_\mu + \frac{4}{\sqrt{1 - \beta_r} \|\Delta_\mu\|^2} \cdot \frac{\|\mu^+\| + \|\mu^-\|}{nk} \right). \quad (A.2.21)$$

Finally, the following holds (w.p. $\geq 1 - 21/n^{20}$):

$$|Q_4| \leq \frac{2\|\mathbf{b}_{r+1}\|\|\mathbf{a}_r\|}{nk}. \qquad (A.2.22)$$

Now, using Lemma A.3 and the assumption about $G_{S_*}^P$, we have (w.p. $\geq 1 - 5/n^{20}$):

$$\|\mathbf{b}_{r+1}\| \leq \sqrt{\beta_{r+1} nk}(\sqrt{d} + C_\psi \sqrt{\beta_{r+1} nk \log nT}) + \beta_{r+1} \gamma nk,$$
$$\|\mathbf{a}_r\| \leq \sqrt{\beta_r nk}(\sqrt{d} + C_\psi \sqrt{\beta_r nk \log nT}) + \beta_r \gamma nk + \sqrt{4nk(d + C_\psi^2 \log n)}. \qquad (A.2.23)$$

Combining (A.2.16), (A.2.20), (A.2.21), (A.2.22), (A.2.23), we have (w.p. $\geq 1 - 25/n^{20}$):

$$\left( \beta_{r+1}(1 - \beta_r)n^2 k^2 - \frac{2(\|\mathbf{b}_{r+1}\| + \beta_{r+1}\|\mathbf{a}_r\|)nk}{\|\Delta_\mu\|} \right) \cdot \Delta_\mu^T C_r^{-1} \Delta_\mu$$

$$- \frac{4(\|\mathbf{b}_{r+1}\| + \beta_{r+1}\|\mathbf{a}_r\|)}{\sqrt{1 - \beta_r}\|\Delta_\mu\|^2}(\|\mu^+\| + \|\mu^-\|) - \frac{2\|\mathbf{b}_{r+1}\|\|\mathbf{a}_r\|}{nk} \leq 0 \quad \text{(A.2.24)}$$

Using (A.2.21), (A.2.23), and the assumption that $\|\Delta_\mu\| \geq 20C_\psi \cdot \gamma \cdot \log(nT)$, $\beta_r \leq 1/(20\gamma C_\psi \log(nT))$, and $n \geq \frac{dmC_\psi^2}{k^2}$, we note that coefficient of $\Delta_\mu^T C_r^{-1} \Delta_\mu$ term above is positive and greater than $\beta_{r+1}n^2 k^2/2$. Lemma now follows by combining (A.2.21), (A.2.23) with assumptions on $\|\Delta_\mu\|$, $\beta_r$ and the above equation. $\qquad \square$

# B    Details of the data sets

| Datasets | Source | #Steps | Feature dimension | # Instances per bag | # Positive instances per bag ($k$) | Positive Examples | | | Negative Examples | | |
|---|---|---|---|---|---|---|---|---|---|---|---|
| | | | | | | Train | Val. | Test | Train | Val. | Test |
| HAR-6 | URL1 | 128 | 9 | 6 | 2 | 6220 | 1132 | 2947 | 0 | 0 | 0 |
| Google-13 | URL2 | 99 | 32 | 4 | 17 | 20883 | 2827 | 2817 | 30205 | 3971 | 4018 |
| STCI-2 | Proprietary | 162 | 32 | 10 | 28 | 32496 | 3921 | 3916 | 10292 | 1302 | 1308 |
| GesturePod-6 | URL3 | 400 | 6 | 3 | 11 | 10154 | 1496 | 198 | 3278 | 1188 | 2354 |
| DSA-19 (SPORTS) | URL4 | 129 | 45 | 4 | 19 | 4560 | 2280 | 2280 | 0 | 0 | 0 |

Table 4: Details of the time series data sets: numbers of time steps in each data point; feature dimension of each data point; and the number of train, test and validation data points for positive and negative classes. The fraction of noisy labels in the positive set is just $1 - k/(\text{\#Instances per bag})$. From the Google-13 dataset, for this study, we use 12 commands as 12 separate *positive* classes and the rest as grouped as *negative* examples. The positive commands are *go, no, on, up, bed, cat, dog, off, one, six, two* and *yes*. The train-test-validate split for DSA-19 is $4 - 2 - 2$ in terms of the number of users. Complete URLs are listed below. Standard splits are used for other datasets.

URL1  `https://archive.ics.uci.edu/ml/datasets/human+activity+recognition+using+smartphones`
URL2  `http://download.tensorflow.org/data/speech_commands_v0.01.tar.gz`
URL3  `https://www.microsoft.com/en-us/research/publication/`
      `gesturepod-programmable-gesture-recognition-augmenting-assistive-devices/`
URL4  `https://archive.ics.uci.edu/ml/datasets/Daily+and+Sports+Activities`

# C    More experiments

(a) GesturePod-6

Figure 5: Variation in accuracy of EMI-RNN for various values of instance width ($\omega$). We infer that $\omega = 200$ is the best instance width for this data set. For datasets where the estimate of signature length ($k$) is unavailable the above graph can be used to pick a good $k$.

(a)

(b)

(c)

Figure 6: Effectiveness of the MI-RNN method in detecting whether there is a zero in a strip of five MNIST digits. We generate and label the training data set – a data point is a positive example if it contains a zero, and negative otherwise. (a) A strip of pixels with five MNIST digits are overlayed, (b) The classifier's confidence at Round 0 of MI-RNN (before label refinement) that the current window contains zero as the strip is rolled past, and (c) The confidence of the output of the MI-RNN method.

Figure 7: In addition to the MIL based approach presented here, an attention mechanism based approach was also explored. There the attempt was to have the attention layer focus only on the class signature thereby facilitating the rejection of the noisy sections of the time series data. Due to the structure in the data (signature being continuous), the hypothesis was that focus of the attention layer would be on a sequence of continuous time steps. Experiments revealed that attention mechanism tends to focus not just on the signature but other aspects as well making extracting the signature difficult. Here a) and b) are representative attention scores obtained on the same task described in Figure 6. It can be seen that the attention layer also focuses on other parts of the input signal along with focusing on the signature (the presence of zero in this case).