[Reviews · NeurIPS 2018]

Reviewer 1



- This paper proposes a multi-instance learning formulation combined with an early prediction technique to address the challenge of fast and efficient classification of sequential data on tiny/resource-constrained devices. The main driving observations include (a) presence of a signature in a small fraction of the data, and (b) that the signatures are predictable using a prefix of the data. The algorithm, EMI-RNN, Early Multi-Instance RNN, shows significant reduction in computation (about 80%) while maintaining/improving the accuracy of the models marginally. - The proposed algorithms E-RNN, and EMI-RNN are based on insights that are described intuitively in the paper. - The gains are impressive and the analysis of the proposed algorithms helps establish usefulness of the approach in real-world deployments of the models. - How sensitive is EMI-RNN to amount of noise? Discussion on gains achieved (accuracy, resources used, and time taken) with respect to percentage of noise in the data should help establish the importance of the approach taken by EMI-RNN. - a thorough analysis of the results with respect to different values of the parameter, $p^{hat}$, added to the appendix, should strengthen the evaluation. - The paper reads well, but another finishing pass should get rid of minor typos: - e.g., line 24 “a wrist bands” —> either “wrist bands” or “a wrist band”. - line 146: “that that” —> that - missing a period, line 118, between “updates” and “They”. - line 143-144-145 doesn’t read well, needs rephrasing. - Notation used for Equation 3.1.1. is explained after an entire paragraph (in a paragraph starting at line 18). This makes reader uncomfortable and causes doubts. Please consider restructuring this part. - Line 281 mentions that Table 2 is provided in the supplement. I understand this decision was taken due to space limitation, but this data is required to establish the appropriateness of the evaluation section. This needs to be included in the main manuscript. - Also, was the conclusions section removed in the interest of space? Please include a concise summary of findings, and next steps. ---------------------- I want to thank the authors for answering my questions; the response helped. I am more convinced of my decision of accepting the paper given that the authors are planning to address the comments in the final version.

Reviewer 2



The authors propose a method for fast and efficient classification of sequential data, motivated by the deployment of such methods on embedded or resource-constrained devices. The method has two key components: (1) it is formulated as a multiple instance learning problem, where each time series is broken down into shorter ones and labels are generated automatically for them, and (2) an early classification formulation that allows the network to emit outputs before processing the whole input sequence. The model is evaluated on five datasets; when compared to the standard LSTM, the proposed solution achieves similar accuracy (sometimes better, sometimes worse) while providing important computational savings. PROS - Efficient model for sequence classification is specific tasks, like wake word detection in voice-driven assistans (e.g. Google Home or Alexa). Speed and low-consumption ML models are key to the development of such systems. - Results on the evaluated datasets are good, as the model can provide computational savings while matching or even improving the accuracy of the baseline LSTM. - The authors show the advantages of the method in resource-constrained devices, namely Raspberry Pi0 and 3. CONS - The model looks somewhat niche, as it seems very tailored for the tasks in the paper. Nevertheless, these tasks are important and interesting to the community. - Although the text is generally well written and motivated, some parts are hard to follow and lack some detail. The manuscript needs a final discussion/conclusion section as well. Please see more extensive comments below. TL;DR: The paper addresses an interesting task and provides motivation for most choices. Some parts of the experimental setup need to be clarified so that readers can understand the contributions more easily and reproduce the results. COMMENTS - The model introduces some new hyperparameters. How sensitive is it to the particular choice of these parameters? Please comment on this, as it is very helpful for practitioners trying to replicate/extend this work. - Equation 3.1.1 is very hard to follow, as most variables have not been defined in the text (e.g. s_i or n). Please clarify this part of the manuscript, it is one of the main contributions of the paper. - What is “train-LSTM”? One SGD step, or a full training? If it is a full training, how many times is this process repeated before convergence? - The description of the evaluation protocol could be more thorough. Why can MI-RNN provide savings when it needs to evaluate some tokens in the time series more than once when the windows overlap, as shown in Figure 1? - There is no description about how the method was implemented in Raspberry Pi. Did the authors use some particular framework? - Please cite the following related paper: Ma et al., “Learning Activity Progression in LSTMs for Activity Detection and Early Detection”, CVPR 2016 - The format of the references section is not consistent. Please select between full conference name or acronym (e.g. [2] or [3]), follow a common format for arxiv pre-prints (e.g. [10] and [31]). Some pre-prints have already been published (e.g. [7]). - Do not attach the main paper with the supplementary material. - There are some typos in the text, please proof-read: 11: tend to (be) discernible 33: predict (the) label 58: We [we] 69: it’s -> its 79: missing comma before “etc” 88: please do not use “etc” and list all the tasks 118: missing whitespace 124-125: use consistent format (with or without comma) for x_{i,T} 144: “models process” 199: are there the same number of positives and negatives (n) in Z, or is this a typo? 213: “noise the bags”? Figure 4: this is a table and should be labeled accordingly ---------------------------------------------------------------------------------------------------------------------------------------------------------------------------------- The results in the paper are good, but I had concerns regarding clarity and reproducibility. The author feedback helped with this, and I believe that the paper will be better once the authors introduce these clarifications in the manuscript. They will also release code so that other can easily reproduce their experiments. After reading the author feedback, I would lean towards acceptance. I increased the overall and reproducibility scores accordingly.

Reviewer 3



Summary This paper proposes a method by using a multiple instance learning formulation along with an early prediction technique to learn a model that can achieve better accuracy. The method is very practical and the ideas of this paper are organized logically. Figures and tables are clear and easy to understand. Also, this paper is written in academic and fluent English. However, there are some unclear parts in the paper. 1. The parameter setting in experiments should be introduced clearly. 2. The equation A.2.18 in supplementary is not easy to understand. Further explanation is encouraged. Several spelling mistakes: 1. Line 58, "We we". 2. Line 64, "typically".

Reviewer 4



To the authors: This is an additional review that I was asked to perform after the rebuttal was submitted by the authors. At this stage, I have not seen the other reviews, the rebuttal or any discussions among reviewers. The paper is concerned with time-series classification on resource-constrained devices. The guiding principle is that for some data modalities it is only necessary to see short time windows in order to make a fairly certain classification. The paper then propose a mechanism for training a sequence classifier that only consider shorter windows; this is then treated as a form of multi-instance learning. Reported results show improved classification performance at notable better speeds. All in all, the paper seem sensible enough to this reviewer, though I have some reservations: *) For the considered data, the problem seem to be that only very large time-windows are labelled, while the actual time-window containing events of interest are short. To me, it seems like the obvious solution would have been to partition that data in smaller time windows ad re-label the data. For many tasks this should be doable, and doing so would have been a reasonable baseline for the experiments. I would strongly encourage the authors to consider this for at least one of their datasets. *) I do not understand the theoretical contribution of the paper. The theorem is not self-contained, e.g. the 'k' is never defined and this seem rather important to see if the results are actually empirically useful. Furthermore, the theorem consider linearly separable data (highly unrealistic) and a loss function that does not match that of the experiments. I understand that theory and practice are often not as well-aligned as one would like, but still I am left to wonder if the theory is useful. *) I found that the first 2.5 pages of the paper could easily be trimmed significantly. The is very little actual content in this part of the paper, with the main focus being on summarizing the contributions. I do not mind such summarizing text (in fact, I like it), but here it's just too much for this reviewer. I strongly encourage the authors to trim this part of the paper. These points aside, the paper seem to provide a reasonable solution to a relevant problem and results appear good. Hence, I lean towards acceptance. After rebuttal (authors can ignore this): NIPS require all reviews to state that the rebuttal was read. In this particular case, the rebuttal was written before the present review, so there is nothing for me to comment on.